



# Estimating sediment thickness by using horizontal distance to outcrop as secondary information

Nils-Otto Kitterød[1,2,3]

[1]Norwegian University of Life Sciences, Environmental Sciences and Natural Resource Management, P.Box 5003, N-1432 Ås, Norway.
[2]Norwegian Institute of Bioeconomy Research, Water Resources, P.Box 115, N-1431 Ås, Norway.
[3]Irstea, UR HHLY, Centre de Lyon-Villeurbanne, 5 rue de la Doua BP 32108, F-69626 Villeurbanne Cedex, France.

*Correspondence to:* Nils-Otto Kitterød (nils-otto.kitterod@nmbu.no)

**Abstract.** Sediment thickness ($D$) was estimated utilizing a publically available well database from Norway, GRANADA. General challenges associated with such databases typically involve clustering and bias of the data material due to preferential sampling. However, if information about horizontal distance to the nearest outcrop ($L$) is included, does the spatial estimation of $D$ improve? This idea was tested comparing two cross-validation results: ordinary kriging ($OK$) where $L$ was disregarded; and co-kriging ($CK$) where cross-covariance between $D$ and $L$ was included. The analysis resulted in only minor differences
between $OK$ and $CK$ in terms of absolute estimation error, however $CK$ produced more precise results than $OK$. All observations were declustered and transformed to standard normal probability density functions before estimation and back transformed for the cross-validation analysis. The semivariogram analysis gave correlation lengths for $D$ and $L$ of approx. 10 km and 6 km. These correlations reduce the estimation variance in the cross-validation analysis because more than 50% of the data material had two or more observations within a radius of 5 km. The small-scale variance of $D$, however, was about 50% of the total
variance, which gave an accuracy of less than 60% for most of the cross-validation cases. Despite of the noisy character in the data material, the analysis demonstrates that $L$ can be used as a secondary information to reduce the estimation variance of $D$.

## 1 Introduction

Global warming and natural climate fluctuations give rise to urgent calls for society to quantify impacts on the hydrological
cycle. These needs are based on numerous indications of expected changes in the pattern of precipitation, temperature and vegetation (Haddeland et al., 2013; Bierkens, 2015; Tang and Oki, 2016). A cardinal question in hydrological modelling is the storage capacity of water in the catchment. Storage capacity determines catchment response to input from rainfall or snow melt events. Storage volumes are therefore important for river discharge calculations and water balance assessments (Meyles et al., 2003; Tromp-van Meerveld and McDonnell, 2006; Beven, 2006; Skaugen et al., 2015). The primary storage capacity in many
catchments is governed by the spatial distribution of sediments in the landscape (Lamb et al., 1997).

Most hydrological models use lumped averages for physical parameters in space, either for large areas or for the entire catchments (Beven and Binley, 1992; Devi et al., 2015). In some of these models, the storage volume is a calibration parameter that may be difficult to assess. In such cases the interpretation of the storage parameter may be misleading or even inconsistent




with physics (Skaugen and Onof, 2013). Thus, to increase prediction reliability, calibration parameters should be replaced by physically based estimates as far as possible.

Soil properties has been registered and mapped by national authorities for many years, but the same attention has not been directed towards the sediment thickness and the bedrock topography. Some remarkable exceptions do exist. Two examples of these include the bedrock topography of Ohio and Iowa Some remarkable exceptions do exist. One examples is the bedrock topography map of Iowa, USA (Witzke et al., 2003). This map was constructed by using well data and digital soil maps that also included observations of outcrops and sparse cover (<1 m) of sediments (Anderson, 2011). In the study presented below, similar data sources were used: a public well database and geological maps showing exposed bedrock and very thin cover of sediments. The intention with this paper is to test simple geostatistical methods to produce similar maps with less consumption of time.

Monitoring of environmental variables takes place as a response to an increasing awareness of human impact on nature. A large number of such variables are available today in public databases. One example is the Norwegian well database GRANADA (NGU, 2016a). According to Norwegian legislation, new wells, boreholes and probe drillings are reported to the Norwegian Geological Survey (Lovdata, 1996). One of the variables stored in GRANADA are recordings of thickness of unconsolidated sediments at the borehole location $D(u_i)$. The purpose of this study is to explore the possibilities of using recordings of $D(u_i)$ to estimate sediment thickness $E[D(u)]$, and estimation variance $Var[D(u)]$. Even though the numbers of recorded $D(u_i)$ increase for every day, the average spatial density of $D(u_i)$ is relatively sparse. Hence, to improve the estimation quality, which in this context means to minimize the estimation variance $Var[D(u)]$, an auxiliary function is attached to $D(u)$, namely the horizontal distance to nearest bedrock outcrop $L(u)$ (Fig. 1).

$L(u)$ is interesting to explore as a secondary variable because it is easy to derive at any location of interest. The statistical relation, however, between $D(u)$ and $L(u)$ is not obvious except when the bedrock is exposed to the atmosphere. If $L(u_j) = 0$, then by definition $D(u_j) = 0$. It does not imply, however, that if $L(u_j)$ is small, $D(u_j)$ is also small, because the bedrock topography may be very irregular or even discontinuous in some places. The contrary is also true: If $L(u_j)$ is large, then $D(u_j)$ is not necessarily always significant. The reason is of course that the bedrock may undulate horizontally below a thin cover of sediments. The working hypothesis, however, is that even though there are numerous exceptions, there might exist a statistical relation between $L(u)$ and $D(u)$ that could be used to reduce the estimation uncertainty of $D(u)$.

It should be emphasized that the relation between $D(u)$ and $L(u)$ depends on the geological setting. The data used for the current study is taken from a location, Norway, where the distribution of unconsolidated sediments is determined by the last glaciation period.

Before presenting the data material in more detail, some statistical challenges should be mentioned. In brief these challenges are related to: asymmetric probability density functions (*pdfs*); clustering; and bias of empirical data.

High resolution environmental data usually deviate strongly from Gaussian *pdfs*. The experimental *pdfs* of $D(u_i)$ and $L(u_i)$ reveal a majority of minor values and a few extremely large values, which imply that they are positively skewed. Standard Gaussian statistics can therefore not be applied directly, at least not without modifications. The challenge of non-Gaussian *pdfs* is relevant for all problems dealing with processes at different scales. Bayesian statistics have given successful contributions



to the estimation of non-Gaussian variables by using Markov Chain Monte Carlo simulation algorithms (MCMC) and by including independent (a priori) information (Omre and Halvorsen, 1989; Andrieu et al., 2003). Recently, an efficient numerical method has been introduced (Rue et al., 2009). In this method the estimation is expressed as a stochastic partial differential equation and the *pdfs* are derived for heterogeneous stochastic fields.

It is beyond the scope of this article to review the big 'Zoo' of different methods, but it should be kept in mind that there exist multitudes of methods that are available for exploring environmental data. Here, in the present case study, the normal score transform was employed (Deutsch and Journel, 1998), which means that after the transform, standard Gaussian statistics were utilized for estimation and afterwards back transformed to the original sampling domain.

Environmental data are prone to preferential sampling. Preferential sampling usually implies clustering and bias. In this
context *clustering* means inhomogeneous sampling frequency in space, while *bias* is systematic over (or under) sampling with respect to low (or high) values. Bias and clustering may appear as independent processes but they may equally well be related to each other by another factor. The data material used for the current study had significant clustering. The reason is simply that wells, boreholes and probe drillings are located where people live. Urban areas account for a higher density of observations than rural or remote areas (Fig. 2). Clustering affects the estimation of statistical moments, and the effect of over- and under
representation of observations should therefore be suppressed. Omre (1984) suggested calculating Thiessen polygons to control clustering effects. The area of the polygons are proportional to the weight coefficients associated to the different observations. In other studies observations are iteratively removed in the calculations of statistical moments (Olea, 2007). For the current study, a grid based method was applied where declustering weights were obtained by gridding the sampling domain. The number of observations within each grid cell were used to calculate weight coefficients (Deutsch and Journel, 1998). In this
way areas with high density of observations received less weight than areas with less frequent observations.

Biased experimental data is ubiquitous in environmental science. A prominent example is observations of precipitation. Several studies document a systematic deficit in the observations due to wind and turbulence (Wolff et al., 2015). In the context of sediment thickness $D(u)$, there are also reasons for systematic underrepresentation of observations with significant $D(u_i)$. In locations where $D(u)$ is large, it is more likely that drilling is terminated because of the drilling costs than in locations with
less sediment thickness. Abandoned wells are not recorded in the database, and the result is a systematic overrepresentation of wells with minor $D(u_i)$. The working hypothesis is to utilize the relation between $D(u)$ and $L(u)$ to improve the estimates of $D(u)$ in a similar way as wind speed is used as secondary information for better estimates of precipitation (Wolff et al., 2015).

## 2   Material

### 2.1   Point observations of sediment thickness

In 1996, Norwegian authorities implemented mandatory reporting of all drillings related to groundwater in mainland Norway (Lovdata, 1996). The purpose of the legislation was to provide the society with relevant groundwater observations. The Geological Survey of Norway (NGU) manages the regulations, and a vital part of this responsibility is the administration of the well database GRANADA. As a public service, the data is freely accessible for downloading (NGU, 2016a). According to recent





statistics about 44% of the recorded boreholes were drilled for the purpose of energy extraction. (NGU, 2016b). At the startup of this study the total number of recorded observations was 54194 (Tab. 1). Of these recordings, 48628 were boreholes, 3740 wells were in unconsolidated sediments, and 1826 were probe drillings. Explicit documentation of $D$ was not available for all GRANADA recordings. For boreholes however, it is possible to derive $D$ indirectly with quite high precision by using data of

the casing length. A casing is necessary in locations with unconsolidated material to prevent sediments from entering the well. Because casing is a significant cost, the casing length is usually reported. Based on the GRANADA recordings, the casing was on average drilled 2 m into the bedrock. Hence, in cases where only casing length was reported, $D$ was set equal to the casing length minus 2 m. In the following, the GRANADA recordings are referred to as boreholes because this is the vast majority of the data material.

**2.2   Land cover information**

The secondary variable: $L$, was calculated from digital maps of unconsolidated sediments (NGU, 2016c). Total areal extension of different sediments are listed in Tab. 1. The sediments are represented in terms of polygons in a Geographical Information System (GIS). Sediments covered by water (lakes, rivers, and glaciers) are not included in Tab. 1. The total sum of land cover polygons is 307104 km$^2$, while the total area of mainland Norway is 323781 km$^2$ ((Kartverket, 2016)). The difference should

in principle be identical to the areal extension of lakes, rivers and glaciers. Thus, according to the land cover polygons (Tab. 1), water covers 5.2% of mainland Norway. Updated figures from the Norwegian Mapping Authority, however, show that lakes (5.7%), glaciers (0.8%) and rivers (0.4%) constitute together 6.9% of mainland Norway ((Kartverket, 2016)). The difference (1.7%) indicate the precision of the total area information and indicate the irreducible uncertainty for this kind of statistics. The relative uncertainty for individual categories is of course higher because positive and negative deviations cancel each other. It

is also important to keep in mind that the actual uncertainty, with respect to areal information, increases with decreasing size of the land category. This precaution is relevant when point information from one data source (GRANADA) is combined with areal information from another source (GIS-maps).

**2.3   Geological setting**

Before explaining the primary screening of boreholes, a few words on the geological setting is necessary. The vast bulk volume

of unconsolidated sediments on mainland Norway is from the last glaciation (Weichselian). More than 90% of the glacial erosion were deposited off shore, and exposed bedrocks or sparse covers of sediments characterize the Norwegian landscape (Olsen et al., 2013). Here, in the current study, the term 'exposed bedrock' includes polygons identified as uncovered bedrock (id. 130, Tab. 1). In addition polygons labeled as 'exposed bedrock or very thin cover of soil or organic matter' were included (id. 100, 101 and 140, Tab. 1). Exposed bedrock constitutes about 35% of mainland Norway according to this definition. Patchy

and thin till material cover about 20% of the land area (id. 12, Tab. 1), and Olsen et al. (2013) include this category when they define areas classified as exposed bedrock. In that case exposed bedrock makes 55% of the land area. Peat bogs cover 5% of the country (id 90, Tab. 1). According to Olsen et al. (2013) the average thickness of the continuous till is approximately 6 m.





They did not include any further discussion on the estimation of sediment thickness based on recorded data. This issue will be elaborated further in the case study presented below.

## 2.4 Data screening

There are no mandatory method for recording of drilling coordinates as part of the GRANADA standard. Quality tags were
therefore attached to the observations to identify the uncertainty of the geographical coordinates. Geographical precision is important to consider during inference on statistical structure of the data material, and it is decisive for spatial resolution of the final estimates. Hence, for the purpose of the current study, observations with less precision than 10 m (18898) were cancelled from further analysis. Wells located on unconsolidated sediments but without any information on $D$, was also omitted (3090) from the analysis. The same was done for probe drillings without information about $D$ (1186 locations). Finally, all boreholes
or probe drillings located inside polygons classified as 'exposed bedrock' (10588) were omitted from further analysis. In these areas $D$ is by definition given as: $E[D|L=0]=0$.

Summing up the cancelled locations (numbers given in parenthesis above), the primary screening reduced the number of recordings from 54194 to 20432. The location of the remaining boreholes ($N=20432$) are indicated in Fig. 2. Some of these boreholes (750) had also recordings of $D=0$, and these wells were also excluded from the statistical analysis.

## 2.5 Exploratory data analysis

Fig. 2 shows that both $D$ and $L$ deviate strongly from Gaussian (normal) probability density functions (*pdfs*). The same is also true for the logarithmic values (Fig. 2). Mean value of D=5.5 m, which corresponds well to the value reported by Olsen et al. (2013), but 50% of the recorded data had $D<=2$ m, which implies a positively skewed *pdf*. Average horizontal distance to outcrop $L=832$ m, while 50% of the boreholes had $L\leq460$ m.

Clustering of boreholes (Fig. 2) can easily be seen on the GRANADA webpage (NGU, 2016a). This uneven spatial sampling affects the inference of statistical moments and the spatial correlation structure.

Mean and standard deviation of $D$ and $L$ as a function of separation distance $h$, is given in Fig. 3 for two different searching windows ($\Delta h=20$ m and $\Delta h=150$ m). It should be noted that the highest values of mean and standard deviation of $D$ occur at small ($h<100$ m) separation distances. This is opposite to what is shown for mean and standard deviations of $L$, which
are small for minor separation distances, increase to maximum values around $h=2.5$ km, and then decay towards $h=10$ km. From Fig. 3 it is clear that when the separation distance $h$ to the nearest borehole increases, the number of low values of $D$ increase. This feature might be caused by preferential sampling, which implies that there is a systematic overrepresentation of drillings that has minor $D$ values. Thus, Fig. 3 indicates a bias in the observations of $D$.

## 3 Method

For the current study, multi-Gaussian methods were applied to estimate sediment thickness $D(u)$, where $u\in\Omega$ and $\Omega$ is the geographical domain covered by the database (in this case mainland Norway). Multi-Gaussian methods are well documented





in the literature (Isaaks and Srivastava, 1989; Journel and Huijbregts, 1989; Deutsch and Journel, 1998), but to make it easier for interested readers to reproduce and improve the results, the most important equations and algorithm are presented in the following. As mentioned above, the main purpose of the study was to evaluate whether the secondary information $L$, can be used to improve the estimates of the primary variable $D$, or not. This question was addressed by performing a conventional cross-validation of the GRANADA boreholes by successively leaving out information on $D$ (but not $L$), and estimate $D$ at the locations where observations of $D$ were left out. First, the cross-validation was performed by including the primary variable $D$ only. Then secondly, the cross-validation was done by including the secondary variable.

More formally expressed, two cumulative density functions ($cdf$) were compared to each other for all borehole locations $u_j$ where $j = 1, ..., N$, and $N$ is the number of GRANADA boreholes (c.f. section above). If the function of interest is Gaussian $Z \in N(0,1)$, then the complete $cdf$ is described by the first two moments. Thus, the task was to compare estimates based on $D$ alone:

$$E[Z_D(u_j)|Z_D(u_i)], \text{ and } Var[Z_D(u_j)|Z_D(u_i)], \tag{1}$$

with estimates based on $D$ and $L$:

$$E[Z_D(u_j)|Z_D(u_i); Z_L(u_j)], \text{ and } Var[Z_D(u_j)|Z_D(u_i); Z_L(u_j)], \tag{2}$$

here, $j = 1, ..., N$, and $i = 1, ..., j-1, i \neq j, j+1, ..., N$, where $N$ is the number of observations (c.f. section 2.4). For this case study ( 1) was obtained by ordinary kriging (*OK*) and ( 2) by co-kriging (*CK*). Before solving ( 1) and ( 2), the experimental data needs preprocessing to suppress effects of preferential sampling, and since Gaussian estimation methods were applied, the data needs to be transformed to a standard normal *pdf*.

### 3.1 Declustering

The purpose of declustering is to compensate for uneven sampling. This was done by giving less weight to observations in areas of high sampling density and a relative increase of weights in areas of sparse sampling. For this case study, the weights were found by gridding of the sampling domain and counting the numbers of observations in each grid cell. The weights were set equal to the inverse of the number of boreholes in the corresponding grid cell. These weights, however, are grid dependent. Hence, the following procedure was implemented to minimize the grid dependency:

1. Decide the size for the grid elements $\Delta u = (\Delta x, \Delta y)$ that constitute a uniform grid.

2. Choose an (arbitrary) origo $u_0$ and make a regular mesh that covers the estimation area ($\Omega$). The mesh consists of $\Delta u_k$ elements, where $k = 1, ..., M$, and $M$ is the number of grid elements.

3. Count the number of boreholes $n_k(u_0)$ and calculate the declustering weights $c_k(u_0)$ for each well in $\Delta u_k$:

$$c_k(u_0) = \frac{1}{n_k(u_0)}, \; k = 1, ..., M, \tag{3}$$

where $M$ is the number of grid elements in the mesh.





4. Because $n_k(u_0)$ ( 3) depends on the grid origo $u_0$ it is necessary to repeat step 2) to 3), and change the grid origo to:

$$u_r = u_0 + r\delta, \tag{4}$$

where $r = 1,...,p$, and the lag $\delta \ll \Delta u$. The number of iterations $p$ should be large enough to get a stable average. Deutsch and Journel (1998) recommend $p \geq 6$. Here, in the current case study $p = 7$, and $\delta = 100$ m.

5. 5. Finally:

$$c_i = \frac{1}{p} \sum_{r=0}^{p} c_k(u_r), \;\; k = 1,...,M, \tag{5}$$

where $c_i$ denotes the declustering weight for the individual boreholes in the database, $i = 1,...,N$, where $N$ is number of boreholes.

The declustering coefficients $c$ ( 5) imply that the total variance of the experimental data is reduced and the correlation length

is increased. This effect is called regularization in the geostatistical terminology. It means that the declustering coefficients also depend on the grid size $\Delta u$. Thus, the final step is to repeat 1) to 5) above, but with a different grid size. The grid size that minimizes the regularization effect should be employed.

## 3.2 Normal score transform

Application of Gaussian interpolation methods imply that the estimated function $Z$ belongs to a standard normal *pdf* $N(0,1)$.

In this case, the stochastic function $X = (D,L)$ is not Gaussian ($\notin N(0,1)$), which means that a transformation is necessary. The normal score transform implies that the quantiles $p_k$ in the original *cdf*, $F(X)$, is corresponding to the quantiles in a standard normal Gaussian *cdf*, G(Z), where $Z \in N(0,1)$ (Goovaerts et al., 2005):

$$Z_X(u_i) = \varphi(X(u_i)) = G^{-1}[F(X(u_i)] = G^{-1}[p_k^*], \tag{6}$$

where $p_k^*$ is the quantiles in the standard normal *cdf*, and $\varphi$ denotes the transformation of $X$ corresponding to the inverse

Gaussian $G^{-1}$ *cdf* of $D$ or $L$. The transformation ( 6) was done by linear interpolation (or extrapolation) from the table of regular sampled $Z \in N(0,1)$ based on the ranked values (percentiles) of $X(u_i) \notin N(0,1)$.

The normal score transform requires a monotonic function to be unique. This is a problem if the data is censored (Huang and Wellner, 1997; Deutsch and Journel, 1998; Goovaerts et al., 2005; Saito and Goovaerts, 2000) which means that the true value are only observed within intervals. This is the case for the lower values in the current experimental data ($D = [0.1; 0.5; 1]$

25 m), which indicate that the true depth is only roughly recorded. For the current study, the normal score transform was done on declustered data which 'corrected' the observations and thus removed over-representation of some observations, thus the transformation to $N(0,1)$ was unique. The back transformation, however, does not reproduce the censored part of the *pdf*.





### 3.3 Experimental semivariogram and cross-semivariogram

The spatial structure of the data $Z$ was described by the experimental semivariogram function:

$$\hat{\gamma}(h) = \frac{1}{2N(h)} \sum_{i=1}^{N(h)} [Z_X(u_i) - Z_X(u_i + h)]^2, \tag{7}$$

where $N(h)$ is the number of data pairs in the separation interval $h$, and where $Z_X$ is the normal score transform ( 6) of either

$D$ or $L$.

In addition to the experimental semivariogram, the mean $m(h)$ and the variance $s^2(h)$ was calculated as a function of $h$:

$$m(h) = \frac{1}{N(h)} \sum_{i=1}^{N(h)} Z_i, \tag{8}$$

and

$$s^2(h) = \frac{1}{N(h)} \sum_{i=1}^{N(h)} (Z_i - m(h))^2 = \frac{1}{N^2(h)} \sum_{i=1}^{N(h)} \sum_{j>i}^{N(h)} (Z_i - Z_j)^2, \tag{9}$$

where $N(h)$ is number of observations for the separation interval $h$.

The experimental cross-semivariogram was estimated by expressing the two functions $Z_D(h)$ and $Z_L(h)$ as a sum of each other:

$$W(h) = Z_D(h) + Z_L(h). \tag{10}$$

This is possible because $D$ and $L$ were sampled in the same locations, and after the normal score transform ( 6) we know by

definition that $E[Z_D] = 0$ and $E[Z_L] = 0$. In that case, the cross-semivariogram can be found by (Myers, 1982):

$$\hat{\gamma}_{Z_D Z_L}(h) = 1/2[\hat{\gamma}_W(h) - \hat{\gamma}_{Z_D}(h) - \hat{\gamma}_{Z_L}(h)], \tag{11}$$

which is valid if $Z_D(h)$ and $Z_L(h)$ are stationary functions in space with finite variance. These properties are difficult to prove in practice, but Myers (1982) suggests that if:

$$|\hat{\gamma}_{Z_D Z_L}(h)| \leq [\hat{\gamma}_{Z_D}(h)\hat{\gamma}_{Z_L}(h)]^{1/2}, \tag{12}$$

then ( 11) is valid.

### 3.4 Semivariogram - and cross-semivariogram maps

Anisotropy structures in the experimental data may be discovered by calculation of semivariogram and cross-semivariogram maps. The same equations ( 7) and ( 11) are applied, but instead of the separation vector $h$ the intrinsic values are calculated as a function of the north-south and east-west components $(h_x, h_y)$ of the separation vector:

$$\hat{\gamma}_{Z_1 Z_2}(h_x, h_y) = \frac{1}{2N(h_x, h_y)} \sum_{i=1}^{N(h_x, h_y)} [Z_1(u_i) - Z_2(u_i + (h_x, h_y))]^2, \tag{13}$$





where $Z_1$ and $Z_2$ denotes stochastic functions. If $Z_1 = Z_2$ (i.e. the normal score transform of $D$ or $L$), then ( 13) is the semivariogram map for $Z_D$ or $Z_L$. If $Z_1 = Z_D$ and $Z_2 = Z_L$, then ( 13) is equivalent to the cross-semivariogram map between $Z_D$ and $Z_L$. The semivariogram (or cross-semivariogram) maps are similar to the experimental semivariogram function, but the semivariance is visualized in terms of a separation matrix instead of a separation vector. By calculating the semivariance in terms of a separation matrix it is possible to reveal large scale (systematic) directional dependencies - called anisotropy. If anisotropy in the observation material is evident, the next step is to calculate directional dependent experimental semivariograms, where the direction of the searching sector is taken from the semivariogram map. The directional dependent properties can be taken into account in the estimation procedure by using the directional dependent searching directions derived from the semivariogram maps. An alternative is to transform the observation coordinates to an isotropic and orthogonal coordinate system (Langsholt et al., 1998).

### 3.5 Semivariogram - and covariance model

The semivariogram model fitted to the experimental semivariogram had the form:

$$\gamma(h) = C_0 + C_1 \left[ 1 - exp\left( \beta \left( \frac{h}{a} \right)^{\alpha} \right) \right], \tag{14}$$

where $C_0$, $C_1$, $a$, and $\alpha$ were the fitting parameters. In geostatistical terms $C_0$ is called the nugget (the variance at $h \rightarrow 0$), $C_0 + C_1$ is the sill (the variance at $h \rightarrow \infty$), $a$ is the range, and $\alpha$ is the exponential coefficient ($1 \leq \alpha \leq 2$). The constant $\beta$ determine the variance at $h = a$. In this case $\beta = -ln(20)$, which is equivalent to 95% of $\gamma(\infty)$. Of that reason $\beta$ is called the practical range in the literature.

The model parameters in ( 14) were obtained by minimizing the objective function $\Upsilon$:

$$\Upsilon(h) = \sum_i |\gamma(h_k) - \hat{\gamma}(h_k)|, \ \ k = 1, ..., K, \tag{15}$$

where $K$ is the number of distance-classes in the semivariogram. For the case study, the objective function was minimized by using the Simulated Annealing Algorithm (MATLAB, 2015).

The kriging equations below are expressed in terms of the covariance function:

$$C(h) = C_0 + C_1 - \gamma(h) = C_1 exp\left( \beta \left( \frac{h}{a} \right)^{\alpha} \right), \tag{16}$$

where the constant $\beta = -ln(20)$, and the parameters $C_0$, $C_1$, $a$, and $\alpha$ were found by minimizing ( 15).

### 3.6 Kriging and co-kriging equations

For this project the kriging and co-kriging equations were implemented in MATLAB (2015), which make it convenient to express the equations in terms of matrix notation. A thorough mathematical derivation of the equations can be found in (Myers, 1982). In matrix notation the estimation is expressed:

$$\hat{\mathbf{Z}} = \mathbf{Z}_{obs} \mathbf{\Lambda}, \tag{17}$$





where $\hat{\mathbf{Z}}$ is the estimated variable in location $u$. If $k = 1, ..., m$ variables are involved, then $\hat{\mathbf{Z}}$ is a row vector with $m$ entries ($1 \times m$ matrix), $\mathbf{Z}_{obs}$ contains the observations in a $1 \times m$ matrix, and $\mathbf{\Lambda}$ is a $m \times m$ matrix where the columns vector are the estimation weights. In this case, $m = 2$, thus in this case ( 17) is written:

$$\left[ \hat{Z}_D, \hat{Z}_L \right] = [\mathbf{Z}_D(obs), \mathbf{Z}_L(obs)] \begin{bmatrix} \Lambda_{DD} & \Lambda_{DL} \\ \Lambda_{LD} & \Lambda_{LL} \end{bmatrix}. \tag{18}$$

For the present case study, the observations: $\mathbf{Z}_D(obs)$, and $\mathbf{Z}_L(obs)$, where available in the same locations $u_i, i = 1, ..., n$. The weights $\Lambda$, are found by solving the kriging equations (Myers, 1982):

$$\mathbf{X} = \mathbf{C}^{-1}\mathbf{C0}, \tag{19}$$

where $\mathbf{C}^{-1}$ denotes the inverse of the matrix $\mathbf{C}$, which in this case reads:

$$\mathbf{C} = \begin{bmatrix} C_{DD}(h) & C_{DL}(h) & I_{DD}^T & I_{DL}^T \\ C_{LD}(h) & C_{LL}(h) & I_{LD}^T & I_{LL}^T \\ I_{DD} & I_{DL} & 0 & 0 \\ I_{LD} & I_{LL} & 0 & 0 \end{bmatrix}, \tag{20}$$

where $C_{kk}(h)$, $k = D, L$ is the covariance model ( 16), $I_{DD} = I_{LL}$ are row vectors of ones ($1 \times n$), and $I_{DL} = I_{LD}$ are row vectors of either ones or zeros, depending on whether all weights should sum up to one or not ($I^T$ indicate the transposed of $I$). The matrix $\mathbf{C0}$ denote the covariance between the point of estimation ($u$) and the observations:

$$\mathbf{C0} = \begin{bmatrix} C0_{DD}(h) & C0_{DL}(h) \\ C0_{LD}(h) & C0_{LL}(h) \\ 1 & 0^* \\ 0^* & 1 \end{bmatrix}, \tag{21}$$

where $C0_{kk}(h)$, $k = D, L$ is given in ( 16). The symbol $0^*$ indicates that the entry might be one or zero, depending on the

Lagrange condition that all weights should sum up to one or only the weights for the single variable estimation problem. Again, zero is the default value. The estimation weights $\Lambda_{kk}(h)$, $k = D, L$ and the Lagrange multipliers $\mu_{kk}(h)$, $k = D, L$, are contained in the $\mathbf{X}$ matrix:

$$\mathbf{X} = \begin{bmatrix} \Lambda_{DD} & \Lambda_{DL} \\ \Lambda_{LD} & \Lambda_{LL} \\ \mu_{DD} & \mu_{DL} \\ \mu_{LD} & \mu_{LL} \end{bmatrix}. \tag{22}$$

The estimation variance $\sigma_K^2$ can be written (Myers, 1982).

$$\sigma_K^2(u) = Var[Z] - \mathbf{X}^T\mathbf{C0}, \tag{23}$$

where the total variance: $Var[Z] = Var[Z_D Z_L]$ for co-kriging and $Var[Z_D]$ for ordinary kriging. The total variance is the sum of the diagonal entries in $C_{DD}(h)$ and $C_{LL}(h)$), where $\mathbf{C0}$ and $\mathbf{X}$ are given in ( 21) and ( 22).





### 3.7 Absolute error, accuracy and precision

The quality of the estimation method depends on the absolute difference between the observed value and the estimated value $(A_E)$:

$$A_E(u_i) = |Z_{obs}(u_i) - \hat{Z}(u_i)|, \tag{24}$$

and mean absolute error $(M_{A_E})$:

$$M_{AE} = \frac{1}{n} \sum_i^n A_E(u_i), \tag{25}$$

and standard deviation of absolute error $(S_{AE})$:

$$S_{AE} = \left( \frac{1}{n-1} \sum_i^n (A_E(u_i) - M_{AE})^2 \right)^{1/2}, \tag{26}$$

where $n$ is the number of cross-validated observations.

In addition, it is necessary to quantify the precision of the estimates. Two concerns are taken into account in this study, first if the estimate is within a given confidence interval $(P_R)$:

$$P_R(u_i) = A_E(u_i) - \omega\sigma_K, \tag{27}$$

where $\omega$ depends on the level of confidence. The accuracy $(A_C)$ of the estimates are then given by:

$$A_C(u_i) = 1 \text{ if } P_R(u_i) \leq 0 \text{ else } A_C(u_i) = 0, \tag{28}$$

and the accuracy is given as a fraction of total numbers of observations $(F_{AC})$:

$$F_{AC} = \frac{1}{n} \sum_i^n A_C(u_i), \tag{29}$$

where $n$ is the number of cross-validated observations.

If two methods have the same level of accuracy, then the method that gives the best precision should be preferred. Precision can be taken into account by scaling the absolute error by the estimation uncertainty:

$$\xi(u_i) = A_E(u_i)/\sigma_K(u_i), \tag{30}$$

and the scaled precision $(S_P)$ is written:

$$S_P(u_i) = \xi(u_i)A_C(u_i), \tag{31}$$

and with the mean scaled precision $(M_{SP})$:

$$M_{SP} = \frac{1}{n} \sum_i^n S_P(u_i), \tag{32}$$





and standard deviation of mean scaled precision ($S_{SP}$):

$$S_{SP} = \left( \frac{1}{n-1} \sum_{i}^{n} \left( S_P(u_i) - M_{SP} \right)^2 \right)^{1/2}, \tag{33}$$

where $n$ is the number of cross-validated observations.

## 4  Results

### 4.1  Declustering and normal score transform

The location of boreholes were clustered in urban areas (Fig. 2). To minimize the impact of this uneven spatial sampling, declustering weights were calculated according to the procedure described above ( 3.1). The window sizes ($w = wx = wy$) applied to calculate the declustering weights, were $w = [500; 1000; 2000; 4000]$ m. Average declustering coefficients were calculated by moving the grid in seven steps $p = 7$, with an offset $\delta = 100$ m ( 4).

The skewness given by the ratio of the median to the mean, for the different declustering windows, $w$, shows that maximum skewness appears for $w = 500$ m (Tab. 2). For $w = 1000$ m, however, the skewness was more similar to the original (raw) observations, thus for the cross-validation analysis the declustering weights were calculated with $w = 1000$ m. The declustering coefficients shows that about 13% of the boreholes had ten or more boreholes located within a neighborhood of 5 km. More than 50% of the boreholes had two or more boreholes within a search radius of 5 km, and about 23% had no other wells within 5 km neighborhood. The normal score transform ( 6) yields per definition a normal *pdf* of the variables involved. The transform relies, however, on the experimental data, which means that sampling of extreme values have impact on the results. The data set used for calculations ($N = 19682$ samples) had a minimum observed $D = 0.05$ m and a maximum $D = 229$ m (Fig. 2). Some of the extreme high values may represent outliers or recording errors, thus for the cross-validation study boreholes with recorded sediment thickness more than 100 m were not included in the calculations. The scatter plot of the raw observations shows the censored character of the data with high frequency of recordings at even numbers (0.10; 0.20; 0.30 m etc.). This is very clear from 0.1 to 1 m, and to some degree between 1 and 10 m (Fig. 4a). After declustering, the artifact was less obvious (Fig. 4b). The semivariogram analysis and the kriging procedures were employed on the normal score data ($Z_D$ and $Z_L$). After kriging, the estimation results were transformed back by inverse normal score transform ( 6) and divided by the declustering coefficients.

### 4.2  Semivariogram maps

Semivariogram maps ( 13) of depth to bedrock $Z_D$, and horizontal distance to outcrop $Z_L$, was calculated to detect large scale anisotropy in the data material. Anisotropy might be identified in Fig. 5 for the range (correlation length) of $Z_D$. The range varies apparently as a function of direction with the slowest decay in the North-West direction (N35W - N45W) and with a somewhat faster decay in the South-East direction. The number of observation pairs had, however, a similar structure, which indicates that the apparent anisotropy might be an artifact due to the clustering of the observations. This presumption was tested





by calculating artificial semivariogram maps based on the same borehole locations but where the observations were substituted by a random number. The artificial semivariogram maps revealed similar structures that can be seen in Fig. 5. Hence, the presumption of an artifact due to clustering cannot be cancelled. For this reason no directional experimental semivariograms were calculated as part of this case study.

## 4.3 Experimental semivariograms and cross-semivariogram

The results of the semivariogram analysis confirm the existence of a correlation structure in the data material (Fig. 6) that might be capitalized when estimating $D(u)$. The model parameters given in Fig. 6 were obtained by minimizing the objective function ( 15) by the Simulated Annealing algorithm (MATLAB, 2015). First, all parameters $[C_0, C_1, a, \alpha]$ were optimized; and then secondly, $C_0$ was fixed and the remaining parameters $[C_1; a; \alpha]$ were simulated. This automatic procedure gave the model parameters shown in Fig. 6. The minimum of the objective function is not well defined everywhere and different combinations of model parameters gave almost similar results. The model parameters in Tab. 3 were evaluated in the cross-validation procedure below. The automatic calibration procedure gave an optimal correlation length of about $a = 10$ km for depth to bedrock $Z_D(h)$ (Fig. 6a). The most prominent feature, however, is the significant nugget value $C_0$, which in this case is about 50% of the total variance: $C_0 + C_1$. The experimental semivariogram for the horizontal distance to outcrop $Z_L(h)$ had minor nugget value compared to the total variance (Fig. 6b). At the same time the correlation length ($a = 5.9$ km) was somewhat shorter compared to $Z_D(h)$. The experimental cross-semivariogram between $Z_D(h)$ and $Z_L(h)$ was calculated according to ( 10) and ( 11). The nugget value was about 10% of the total variance in this case with a correlation length of $a = 2.7$ km. Finally, the cross-semivariogram was tested according to ( 12), but none of the parameter combinations in Tab. 3 violated the criteria.

## 4.4 Cross-validation

The purpose of the cross-validation was to evaluate the impact of using horizontal distance to outcrop as an additional variable for estimation of sediment thickness above the bedrock. In this case, the cross-validation was performed by leaving one observation out. At the point where the observed value was left out, ordinary kriging (*OK*) and co-kriging (*CK*) were performed by using the global model parameters given in Tab. 3. The differences between the estimation results and the observations left out, were used to quantify the quality of the estimation procedure. Three criteria were used to distinguish the two estimation procedures: the absolute error ( 24 and 25); the accuracy of the estimation results ( 28 and 29); and the precision of the estimation results ( 31 and 32).

In general, both *OK* and *CK* over estimate minor depths to bedrock and underestimate significant depths (Fig. 7). The most important estimation criteria is usually considered to be the absolute error ( 24) and the mean of the absolute error ( 25). With the model parameters tested in Tab. 3 there are no significant differences in the mean absolute error ( 25) between the *OK* and *CK* estimates (Tab. 4). The *CK* estimates have slightly lower mean absolute error than the *OK* estimates unless the nugget value ($C_0$) for the cross-covariance between $D$ and $L$ approaches half of the total variance: $C_0 + C_1$ (Tab. 4).




In cases with no significant difference in absolute error, accuracy and precision are distinguished as the superior method. For this case study, the definition of accuracy ( 28) and scaled precision ( 31) are related to the estimation variance ( 23), and in this respect, *CK* is superior (Fig. 8).

In Fig. 9 scaled precision ( 31) is sorted and given as a function of cumulative accuracy $S_{AC}$:

$$S_{AC} = \Sigma_i^j A_C(u_i)/n_{max}, \quad j = 1, ..., n_{max}, \tag{34}$$

where $n_{max}$ is the number of estimates where $A_C = 1$.

As long as the absolute estimation errors ( 24) are similar, *OK* yields higher accuracy than *CK* because *CK* has lower estimation variance. This result follows directly from the definitions ( 27 and  28). With $\omega = 1$ in ( 27), the *OK* estimates gave an accuracy from 60 to 65%, while *CK* had accuracy of 50-60%. At the same time *CK* yields an overall higher precision than *OK* because of lower estimation variances (Fig. 9).

A final result that deserves some attention is the location of estimates that did or did not fulfill the accuracy criteria. In Fig. 10 this is illustrated for mainland Norway and in Fig. 11 for the Oslo area. Three categories were visualized: Locations with low accuracy ($AC = 0,\ 28$); locations with good accuracy ($AC = 1$, 28 obtained either by *OK* or *CK*; and locations with $AC = 1$ obtained only by the *CK* method. For all cases $\omega = 1$ ( 27).

## 5 Discussion

Attention has been directed towards sediment thickness, $D$, in this article. The question has been raised whether information derived from public well databases on $D(u_i)$ can be utilized for continuous estimation of $D(u)$. A motivation for this attention has been the potential application of spatial estimates of $D(u)$ in hydrology and geo-engineering. Combined with available information on soil properties or digital terrain elevation, storage capacity of water or bedrock topography might be estimated within predefined uncertainties and with feasible resources. It should be emphasized, however, that the purpose of the application should be taken into account when choosing the estimation method. In this case study, the normal score transforms and Gaussian estimation methods were applied, but none of these methods provide robust estimates of extreme values. If for example, maximum $D(u)$ is an important issue, stochastic simulation or non-Gaussian methods should be taken into account. Such topics, however, are left for further studies.

### 5.1 Clustering and bias

For the current case study, $D(u)$ was derived from an open access well database (NGU, 2016a). Public databases are prone to preferential sampling. In this context, preferential sampling implies two specific challenges that need to be discussed, namely clustering and bias. Clustering is due to the fact that wells and boreholes are located where people need them, thus the spatial frequency of boreholes mirror the population density (Fig. 2). Clustering of observations have impact on statistical inference regarding statistical moments and semivariograms. Different approaches have been suggested to control the clustering effects. Olea (2007) suggested removal of wells randomly in areas with high density of observations, and then calculate experimen-





tal semivariograms based on the remaining observations. The experimental semivariograms were sensitive to the size of the searching window where clustered observations were removed. Thus, this method was disregarded in the current case study because the algorithm did not yield robust results.

Omre (1984) suggested controlling clustering effects in the semivariogram by calculating weights that are inversely propor-
5 tional to the Thiessen polygons for each observation. This method provides a set of weights that are mathematically sound, but it is relatively expensive with respect to computer resources especially if the number of observations are large. Instead of Thiessen polygons a less computer demanding algorithm was employed, namely the moving grid method (Deutsch and Journel, 1998). By this method the declustering weights are inverse proportional to the average number of observations within the moving window ( 3.1). The declustering weights depend on the size of the window (Tab. 2). In general it is recommended to
10 use the window size $w$, that maximizes the skewness of the *pdf(s)*, which in this case was $w = 500$ m. However, $w = 1000$ m gave a skewness for $D(u)$ that was closer to the original data, thus the semivariograms was based on a declustering window $w = 1000$ m. The mean value from raw (not declustered) data was 5.5 m, but the declustered mean was reduced to 3.4 m and 2.8 m for $w = 500$ and $w = 1000$ m respectively (Tab. 2).

The problem of biased recordings of $D(u)$ in the database is more difficult to assess. There are good reasons to expect that
bias exists and that minor sediment thicknesses are over represented in the database. One indication is that mean and standard deviation are highest at minor separation distances, which indicate that willingness to continue drilling is less if $D(u)$ is large, and if there are no other wells in the close neighborhood (Fig. 3).

Biased observations are a common problem for datasets sampled in open large-scale environments. The impact of bias may be controlled if there exist independent information on processes related to the variable of interest. Goovaerts et al. (2005) did
a case study based on biased observations of arsenic concentration in groundwater. They used geological maps and utilized knowledge of arsenic concentration in specific geological units to control the bias. Wolff et al. (2015) reported biased recordings of precipitation from a meteorological gauge station. In this case the bias was due to turbulence in the wind field around the gauge equipment. They recorded wind speed and temperature together with precipitation and other meteorological variables, and derived functions for bias correction by application of Bayesian statistics. By similar token, horizontal distance to outcrop
$L(u)$ was evaluated as secondary information to estimate sediment thickness $D(u)$ by using a data set that most likely is prone to bias.

The cross-validation exercise presented here, cannot verify a general relation between $D(u)$ and $L(u)$, but the results show that the estimation uncertainty is reduced by using $L(u)$ as a secondary function. Non-biased relations between $D(u)$ and $L(u)$ ought to be investigated by further research for example by utilizing dataset from geotechnical probe drillings. Results from
30 such studies would increase the value of the GRANADA database and other similar databases.

## 5.2 Cross-validation

The cross-validation analysis indicate low estimation accuracy in urban areas. One reason for this result might be anthropogenic reallocation of unconsolidated matter, which includes removal of sediments in some places and deposition of unconsolidated matter in others. Similar problems might also be valid for identification of horizontal distance to outcrop. For further studies





such locations might be disregarded or given less weights. One option is to allocate a quality tag to the $D(u)$ recordings in the same manner as done for recordings of geographical coordinates.

Both *OK* and *CK* overestimates small $D(u)$ and underestimate large $D(u)$ (Fig. 7). This result is typical for Gaussian estimation methods applied on observations with positively skewed *pdfs*. Other case studies report similar results (Goovaerts
et al., 2005), but it should be noticed that the double logarithmic scale exaggerates the deviations especially for minor depths.

The observations of $D(u)$ had a high fraction of small scale noise ($C_0$ in Fig. 6) relative to the total variance: $C_0 + C_1$ (Fig. 6). Efforts should be taken to control $C_0$. One abatement measure might be achieved by attaching a quality assurance tag to $D(u)$. In this way low quality recordings could receive less weights or be filtered out. This kind of measures would increase the quality of the GRANADA database.

Despite of these uncertainties the cross-validation shows that the accuracy is higher than 60% for the model parameters with highest scores (Tab. 4). For this case study, the estimation accuracy was set equal to one if the absolute estimation error was less than one standard deviation of the estimation uncertainty and zero for all others ( 27 and  28). By this definition, the accuracy increases by increasing estimation variance, which means that accuracy should be evaluated together with the estimation variance ( 23; Fig. 8). For stochastic simulation the precision of the estimates is of primary interest. In such cases, the probability
of extreme realizations may also be quantified. For such applications, the precision is more important than the accuracy of the estimation method. The cross-validation results show that the precision in general is higher if the horizontal distance to the outcrop$L(u)$ was included (Fig. 9). Because precision increases as a function of decreasing estimation variance, the cross-validations show that $L(u)$ should be included despite of the significant uncertainties in the experimental data material.

## 5.3 Further studies

These results indicate that more advanced estimation procedures should be considered. In this case study, the total estimation domain (mainland Norway) is considered as homogeneous with respect to variance and correlation length. Methods that take local model parameters and local anisotropy into account may reduce the absolute estimation error but not necessarily the estimation variance. The same is true with respect to estimation methods that are more robust with respect to estimation of extreme realizations. For estimation of most likely minimum and maximum thickness of sediments within a given estimation
area, stochastic simulations are recommended.

After initiation of this case study, the number of recorded boreholes, wells, and probe drillings in the GRANADA database have increased significantly (NGU, 2016b). The new recordings might be used as an independent dataset for cross-validation purposes. One interesting candidate for further work is the approach suggested by Rue et al. (2009). They approximate the estimation problem to stochastic partial differential equations. In this method non-stationarity of statistical moments are taken
into account, and at the same time less computer resources are spent on matrix inversions which is a challenge for applications with a large number of observations. (Lindgren et al., 2011; Ingebrigtsen et al., 2014; Hu and Steinsland, 2016).

Finally, it is appropriate to recall that the primary purpose of the GRANADA database is not the recording of sediment thickness $D(u)$ alone, but to provide information on groundwater resources in general (Lovdata, 1996). The present article should be read in light of this purpose, which is a call to increase research on data from the GRANADA database and other





public databases. Drilling of boreholes concerns the society and should not be regarded as a private issue even though the borehole and the wells are located on private property. This is quite evident for urban areas where subsurface infrastructure is vital for transport; communications; and other community services. It is also evident that wells and boreholes affect the common society if extraction of water and/or energy exceeds sustainable levels. Without legislation requiring the registration
of individual boreholes and wells, it is very difficult to monitor and implement assessments strategies that promote a common resource while minimizing negative consequences for nearest neighbors; downstream communities; or vulnerable ecosystems.

## 6   Summary and conclusions

The open access database GRANADA (NGU, 2016a)) were used to derive point recordings of thickness of sediment above the bedrock $D(u)$. For each $D(u)$ horizontal distance to nearest outcrop $L(u)$ was derived from geological maps. The purpose
was to utilize $L(u)$ as a secondary function for estimation of $D(u)$. Two estimation methods were employed: ordinary kriging ($OK$) and co-kriging ($CK$). A cross-validation analysis was performed to evaluate the additional information in the secondary function $L(u)$. $L(u)$ was disregarded in $OK$-estimation but included in $CK$-estimation. The cross-validation results showed that $CK$ provided overall lower mean absolute error compared to the $OK$ results, but the differences was minor. The estimation uncertainty determines the estimation accuracy and the precision. These quantities might be considered as equally important
as the mean absolute error. With respect to the estimation precision, the $CK$ estimates were superior to $OK$ estimates, demonstrating the value of using $L(u)$ as an secondary function for estimation of $D(u)$. The problem of clustering of observations can be controlled by calculation of declustering weights, but the relation between $D(u)$ and $L(u)$ should be explored in further studies to control the effect of biased observations.

The semivariogram analysis revealed a correlation length (range) for $D$ of approximately 10 km and about 6 km for $L$. The
cross-semivariogram between $D$ and $L$ gave a corresponding length of 2.7 km (Fig. 6). The recordings of $D$ had a significant small-scale variance (nugget value). Despite of this, the estimation accuracy was quite high (Tab. 4). Between 50% and 60% of the cross-validation recordings had an accuracy of less than one kriging error $\sigma_K$ ( 23). Based on these results, continuous estimates of $D(u)$ can be derived for mainland Norway.

*Acknowledgements.* To the Geological Survey of Norway (NGU) for providing the dataset; to Bioforsk (now NIBIO) for GIS assistance;
and to Camille Jouin who carefully prepared and documented the data.





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

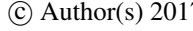


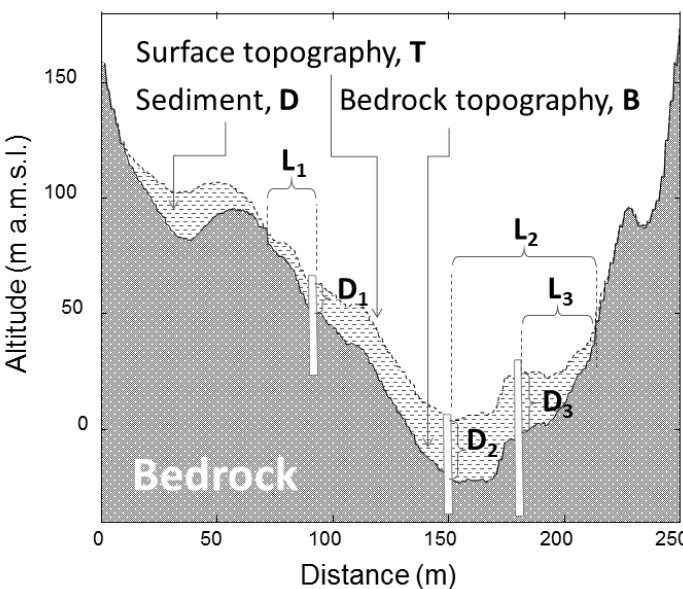

**Figure 1.** Surface topography $T$, sediment thickness $D$, and bedrock topography $B$. Observations of $D_i$ are indicated in three boreholes ($i = 1, 2, 3$) and with the associated horizontal distance to nearest outcrop ($L_i$). In areas where $B$ is not exposed, $B$ can be estimated by using observations of $D$ as primary variable and information of $L$ as secondary information.



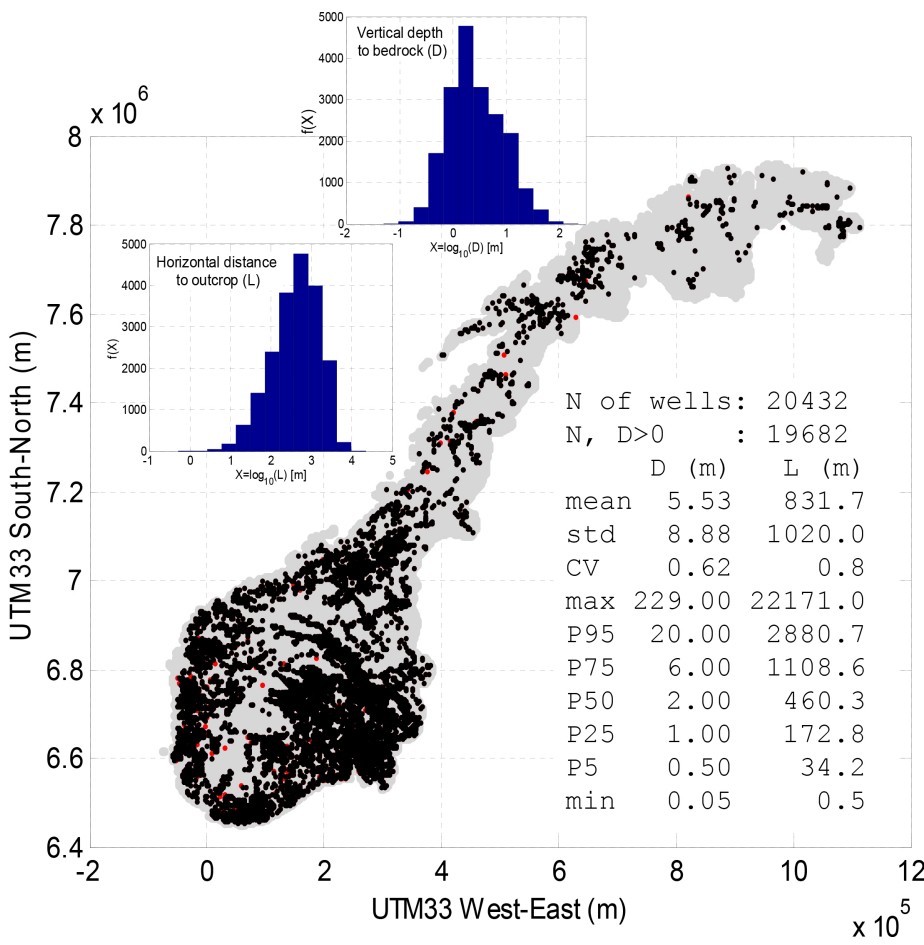

**Figure 2.** Location of wells, boreholes and probe drillings in the GRANADA database (NGU, 2016a). 20432 number of observations (N) were included in this study (see text for screening of observations). Black dots indicate locations where sediment thickness $D(u_i) > 0$, $i = 1, ..., M$, $M = 19682$. Red dots indicate locations where $D(u_j) = 0$, $j = 1, ..., K$, $K = N - M$. Horizontal distance to nearest outcrop $L(u_i)$, were calculated for locations where $D(u_i) > 0$. Histograms of $log_{10}(D|D > 0)$ and $log_{10}(L|D > 0)$ indicate significant deviation from normal probability density functions (lower right corner). Statistical parameters and percentiles for $D$ and $L$ are given in upper left corner.





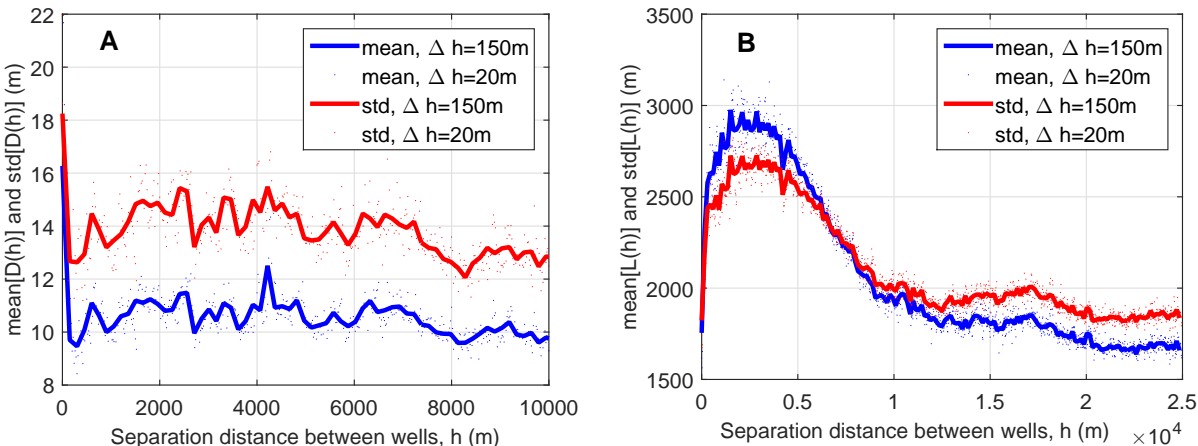

**Figure 3.** Mean and standard deviation as a function of separation distance $h$ (m), sediment thickness $D$ to the right (A), and horizontal distance to outcrop $L$ to the left (B).

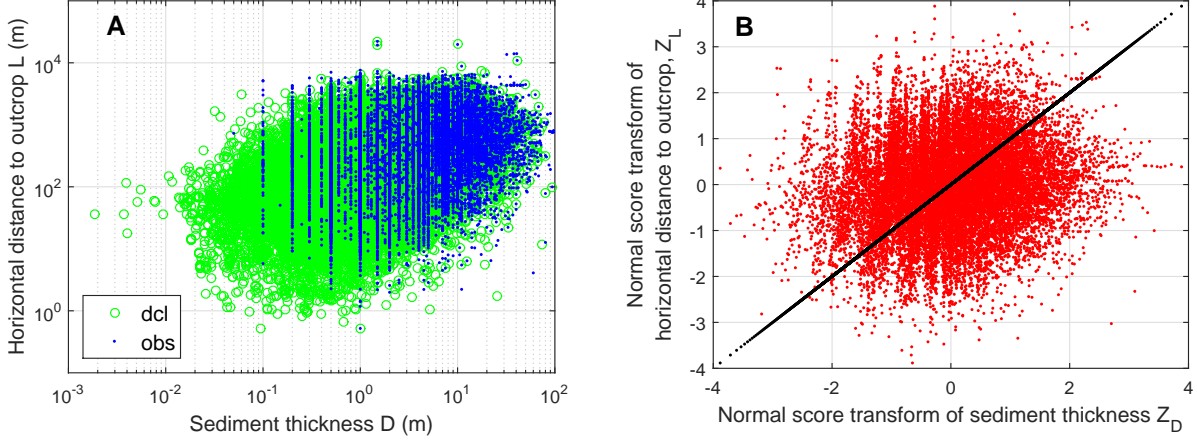

**Figure 4.** Scatter plot of sediment thickness $D$ and horizontal distance to outcrop $L$. Original (obs) and declustered (dcl) observations to the left (A), and normal score transforms $Z_D$ and $Z_L$ to the right (B). Black line indicate a 'perfect' (1:1) relation between $Z_D$ and $Z_L$.





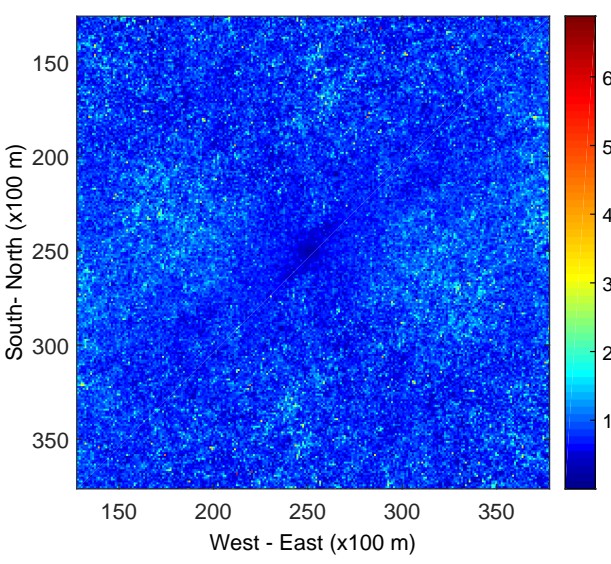

**Figure 5.** Semivariogram map ($\hat{\gamma}(h_x, h_y)$) of normal score transformed sediment thickness $Z_D$, grid cells of $100 \times 100$ m.





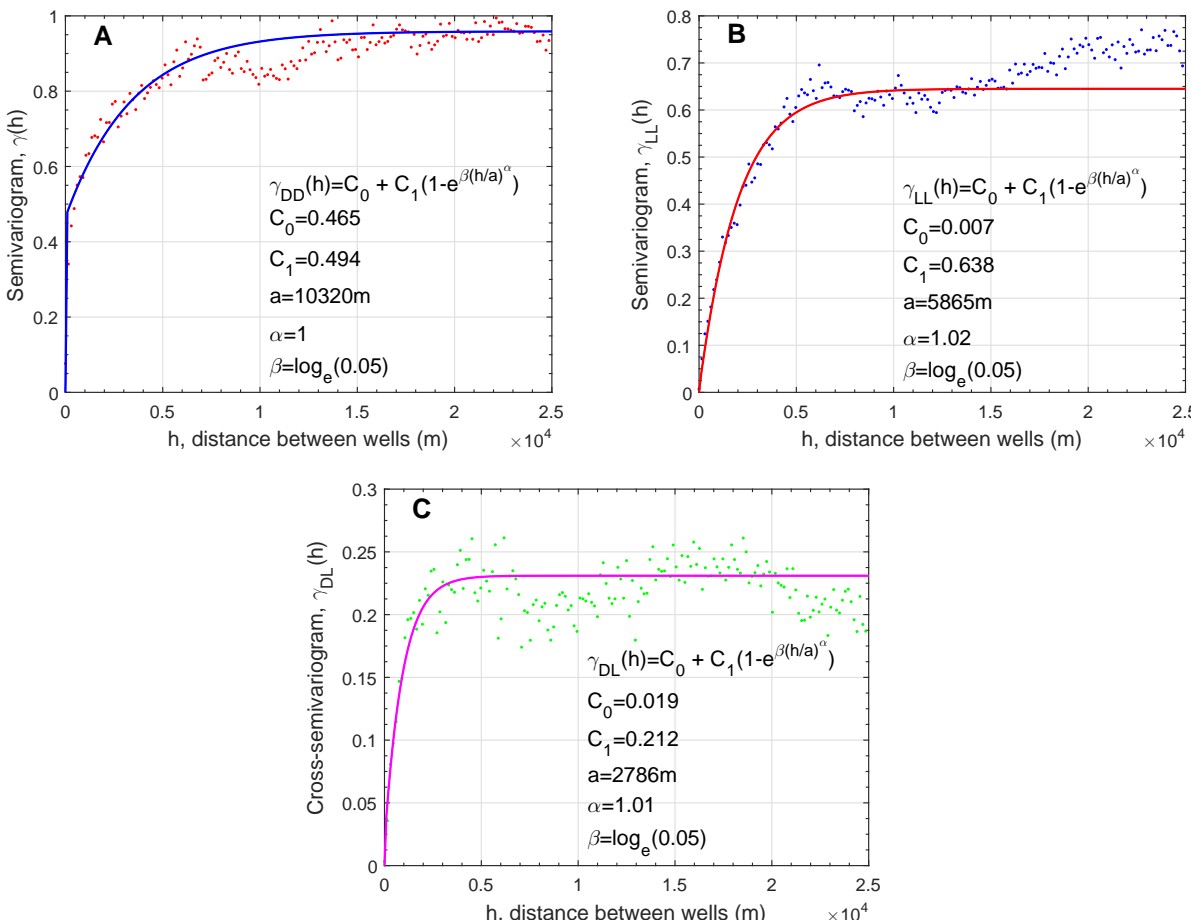

**Figure 6.** Semivariogram and cross-semivariogram functions for normal score data: Semivariograms for sediment thickness $Z_D$ (A); and horisontal distance to nearest outcrop $Z_L$ (B). Cross-semivariongram for $Z_D$ and $Z_L$ (C). Dots indicate the experimental data and soild lines are the model functions.





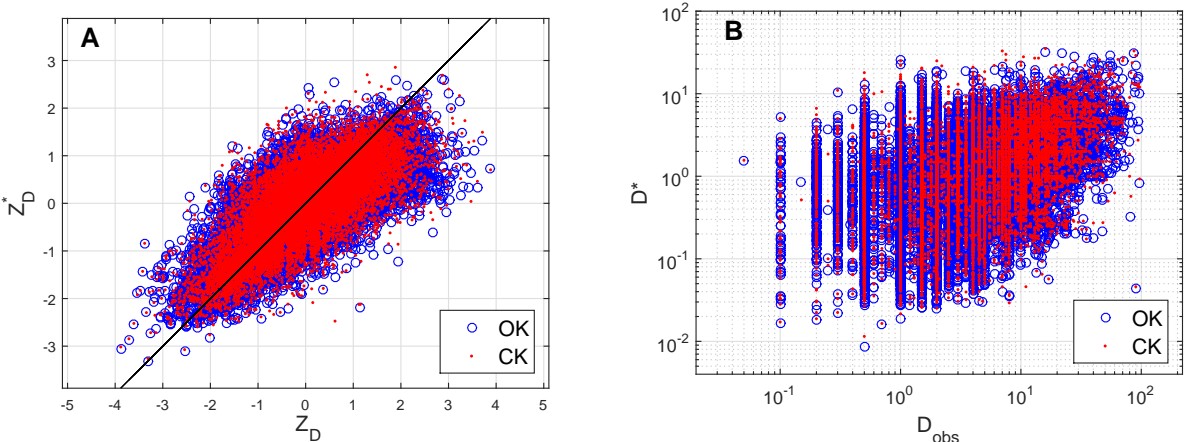

**Figure 7.** Cross-validation results of sediment thickness $D$. Normal score observations $Z_D$ against estimates $Z_D^*$ to the left (A), and raw observations $D$ and estimates $D^*$ to the right (B). *OK* denote results from ordinary kriging, and *CK* from co-kriging.

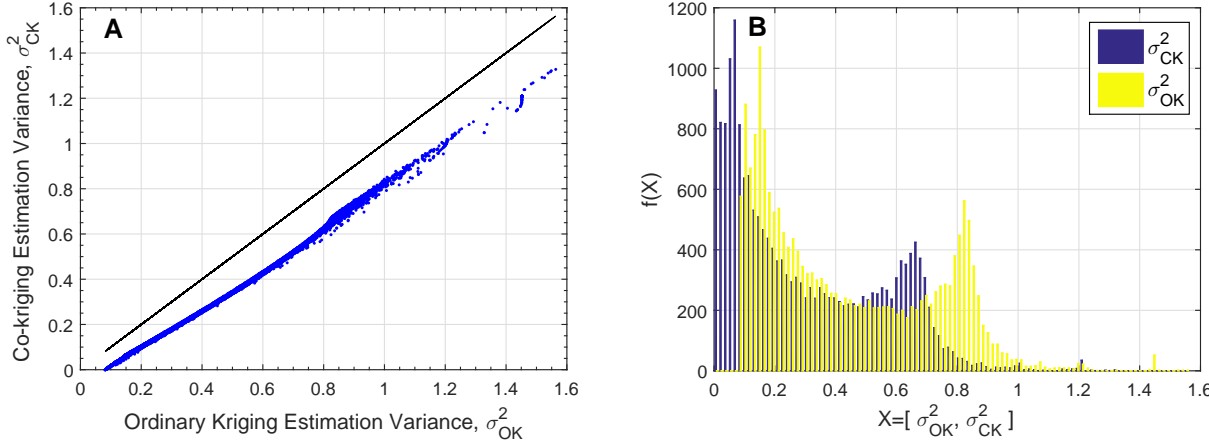

**Figure 8.** Estimation variance (eq. 23) for case $F$ (Tab.3 and Tab.4), with ordinary kriging *OK* and co-kriging *CK* results. *CK* estimation variance are lower than *OK* estimation variance from (A). Black line indicate a 1:1 relation. The histograms to the right (B) shows the estimation variance for *OK* and *CK*.




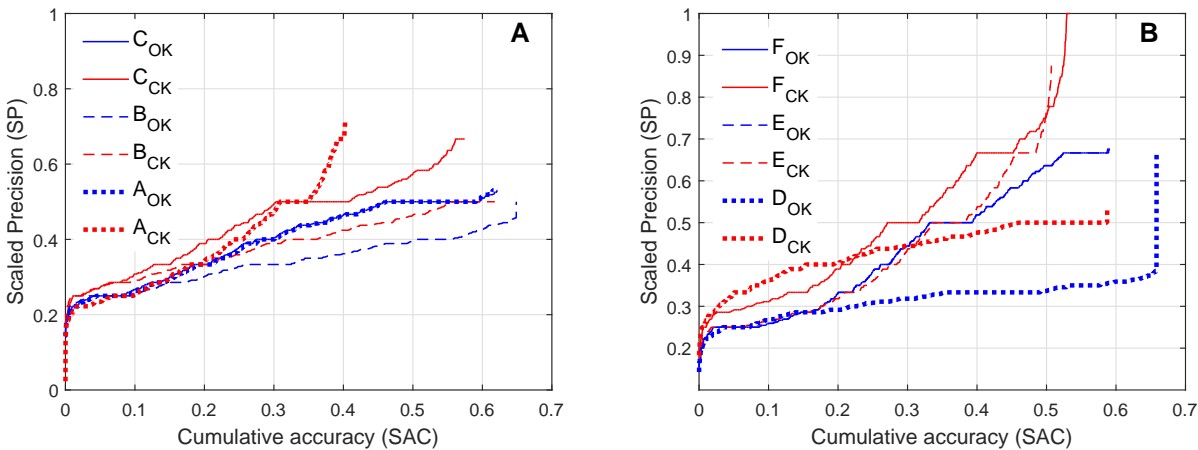

**Figure 9.** Scaled precision ( 31) plotted as a function of cumulative accuracy (34) for estimation cases $[A; B; C]$ to the left (A), and $[D; E; F]$ to the right (B). Model parameters are given in Tab.3. Ordinary kriging (*OK*) yields highest accuracy for most cases, but co-kriging *CK* gave overall highest precision.

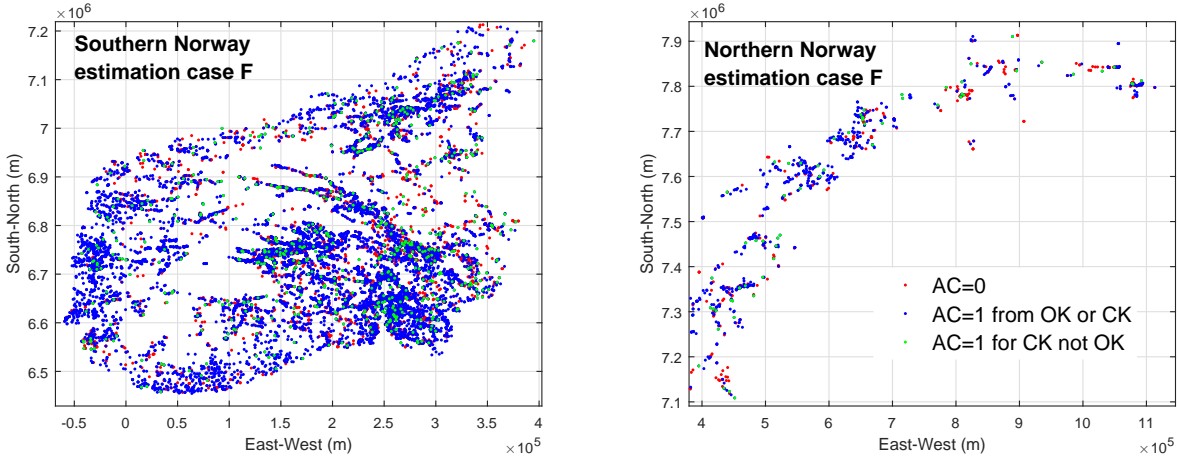

**Figure 10.** Cross-validation results of Granada boreholes (NGU, 2016a) for estimation case F (Tab.3 and Tab.4). For this case, 37% of the locations did not fullfill the accuracy criteria (28) indicated by red dots; 63% of the locatios did fulfill the accuracy criteria by either the *OK* or the *CK* method (blue dots); for 3.5% of the locations the accuracy criteria was fullfilled by the *CK* method and not the *OK* method (green dots). For 9.2% of the locations the accuracy criteria was met by the *OK* method only (not shown). Geographical coordinates are given for UTM zone 33.



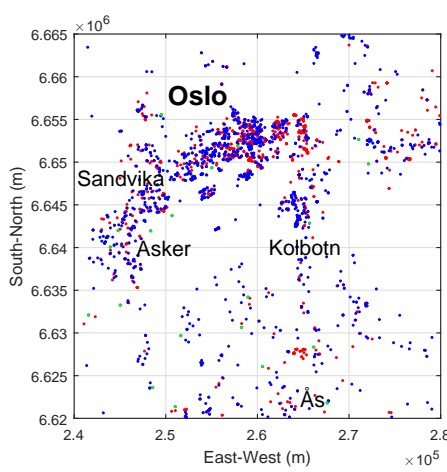

**Figure 11.** Cross-validation results of GRANADA boreholes (NGU, 2016a) for the Oslo area for cross-validation case $F$ (Tab.3 and Tab.4). Red dots indicate low accuracy $AC = 0$ (28), blue dots are locations with $AC = 1$ obtained with ordinary kriging *OK*, or co-kriging *CK*. Green dots indicate boreholes where $AC = 1$ for *CK* only.





**Table 1.** Land cover statistics of mainland Norway

| Id[a] | Land cover | $A_i^b$ (km$^2$) | N$^c$ | $F_{atm}^d$ (%) | $F_{tot}^e$ (%) |
|---|---|---|---|---|---|
| 130 | Exposed bedrock | 97000 | 9562 | 31.59 | 29.96 |
| 12[f] | Till material, patchy or thin cover over bedrock | 80719 | 10311 | 26.28 | 24.93 |
| 11[g] | Till material, continuous cover, great thickness locally | 65008 | 10640 | 21.17 | 20.08 |
| 90 | Peat bogs and swamps | 17000 | 1445 | 5.54 | 5.25 |
| 70[h] | Weathered deposits, not divided by thickness | 15600 | 3464 | 5.08 | 4.82 |
| 20[i] | Fluvial sediments | 8829 | 6095 | 2.87 | 2.73 |
| 41[j] | Marine and coastal sediments, coherent, often great thickness | 7600 | 5932 | 1.56 | 1.48 |
| 81[k] | Avalanche materials and landslides | 7272 | 235 | 2.37 | 2.25 |
| 43[l] | Marine, beach sediments, patchy or thin cover over bedrock | 2676 | 3625 | 0.87 | 0.83 |
| 14 | Till modified by running water (ablation moraine) | 1900 | 67 | 0.62 | 0.59 |
| 21[m] | Glaciofluvial sediments | 1769 | 683 | 0.58 | 0.55 |
| 15 | Ice-marginal deposits | 1000 | 264 | 0.33 | 0.31 |
| 120 | Anthropogenic deposits, unspecified | 350 | 1650 | 0.11 | 0.11 |
| 30[n] | Glaciolacustrine and lake sediments | 253 | 175 | 0.082 | 0.078 |
| 60 | Eolian (Wind) sediments | 100 | 46 | 0.033 | 0.031 |
| 88[o] | Scree, clay slides, rock falls etc. | 28 | 0 | 0.0091 | 0.0086 |
| | Sum | 307104 | 54194 | 100.00 | 94.85 |

[a]) Land cover identificantion numbers (NGU, 2016d).

[b]) Area of land cover polygons exposed to the atmosphere, $A_{atm} = \sum A_i$ =307104 km$^2$. The total area of mainland Norway is: $A_{tot}$ = 323781 km$^2$ (Kartverket, 2016).

[c]) Number of recorded boreholes, wells, and probe drillings in GRANADA 2010 (NGU, 2016d).

[d]) Fraction of land cover polygons relative to $A_{atm}$.

[e]) Fraction of land cover polygons relative to $A_{tot}$. Mainland Norway covered by water:$1 - A_{atm}/A_{tot}$ =1-0.9485=0.0514.

[f]) Includes id. 12 (65000 km$^2$), 100 (thin humus cover, 12000 km$^2$), 140 (3500 km$^2$), 101 (210 km$^2$), 10 (5.8 km$^2$) and 13 (3.5 km$^2$).

[g]) Includes id. 11 (65000 km$^2$) id. 16 (drumlin, 8 km$^2$).

[h]) Includes id. 70 (7000 km$^2$), 73 (5100 km$^2$), 71 (2300 km$^2$) and 72 (1200 km$^2$).

[i]) Includes id. 20 (4700 km$^2$), 50 (4000 km$^2$), 54 (2600 km$^2$), 55 (76 km$^2$).

[j]) Includes id. 41 (4800 km$^2$), 42 (2800 km$^2$).

[k]) Includes id. 81 (5000 km$^2$), 82 (2200 km$^2$), 80 (69 km$^2$) and 301 (2.5 km$^2$).

[l]) Includes id. 43 (2600 km$^2$), 40 (76 km$^2$).

[m]) Includes id. 21 (1700 km$^2$), 22 (69 km$^2$).

[n]) Includes id. 30 (190 km$^2$), 36 (38 km$^2$), 35 (25 km$^2$).

[o]) Includes id. 88 (scree, 17 km$^2$), 307, 102, 1, 122, 31, 304, 308, 313, 315, 53 and 316.





**Table 2.** Median and mean of depth to bedrock $D$ (m), and horizontal distance to outcrop $L$ (m), for raw observations (window size = 0 m) and declustered data with window $[500; 1000; 2000; 4000]$ m. The skewness index, skw=median/mean.

| window size (m) | 0 | 500 | 1000 | 2000 | 4000 |
|---|---|---|---|---|---|
| D median | 2.000 | 1.286 | 1.000 | 0.594 | 0.321 |
| D mean | 5.451 | 3.394 | 2.770 | 2.043 | 1.316 |
| D skw | 0.367 | 0.379 | 0.361 | 0.291 | 0.244 |
| L median | 458.63 | 227.94 | 156.79 | 91.64 | 47.67 |
| L mean | 827.46 | 491.69 | 382.14 | 268.67 | 169.06 |
| L skw | 0.554 | 0.464 | 0.410 | 0.341 | 0.282 |





**Table 3.** Covariance and cross-covariance model parameters[a] (16) used for cross-validation.

| Case | | $C_0$ | $C_1$ | $a$ | $\alpha$ |
|---|---|---|---|---|---|
| A | $C_{DD}$ | 2.09e-01 | 6.72e-01 | 4.478e+03 | 1.00 |
| | $C_{DL}$ | 1.00e-01 | 1.10e-01 | 1.512e+03 | 1.65 |
| | $C_{LD}$ | 1.00e-01 | 1.10e-01 | 1.512e+03 | 1.65 |
| | $C_{LL}$ | 0.39e-01 | 6.06e-01 | 6.049e+03 | 1.05 |
| B | $C_{DD}$ | 3.44e-01 | 5.18e-01 | 4.380e+03 | 1.00 |
| | $C_{DL}$ | 0.00e+00 | 1.00e-01 | 1.512e+03 | 1.65 |
| | $C_{LD}$ | 0.00e+00 | 1.00e-01 | 1.512e+03 | 1.65 |
| | $C_{LL}$ | 0.39e-01 | 6.06e-01 | 6.049e+03 | 1.05 |
| C | $C_{DD}$ | 2.09e-01 | 6.72e-01 | 4.478e+03 | 1.00 |
| | $C_{DL}$ | 0.00e+00 | 1.00e-01 | 1.512e+03 | 1.65 |
| | $C_{LD}$ | 0.00e+00 | 1.00e-01 | 1.512e+03 | 1.65 |
| | $C_{LL}$ | 0.39e-01 | 6.06e-01 | 6.049e+03 | 1.05 |

| Case | | $C_0$ | $C_1$ | $a$ | $\alpha$ |
|---|---|---|---|---|---|
| D | $C_{DD}$ | 4.65e-01 | 4.94e-01 | 10.320e+03 | 1.00 |
| | $C_{DL}$ | 1.90e-02 | 2.12e-01 | 2.786e+03 | 1.01 |
| | $C_{LD}$ | 1.90e-02 | 2.12e-01 | 2.786e+03 | 1.01 |
| | $C_{LL}$ | 7.00e-03 | 6.38e-01 | 5.865e+03 | 1.02 |
| E | $C_{DD}$ | 7.70e-02 | 7.26e-01 | 2.371e+03 | 1.00 |
| | $C_{DL}$ | 1.90e-02 | 2.12e-01 | 2.786e+03 | 1.01 |
| | $C_{LD}$ | 1.90e-02 | 2.12e-01 | 2.786e+03 | 1.01 |
| | $C_{LL}$ | 7.00e-03 | 6.38e-01 | 5.865e+03 | 1.02 |
| F | $C_{DD}$ | 7.70e-02 | 7.26e-01 | 2.371e+03 | 1.00 |
| | $C_{DL}$ | 3.00e-03 | 2.07e-01 | 2.402e+03 | 1.01 |
| | $C_{LD}$ | 3.00e-03 | 2.07e-01 | 2.402e+03 | 1.01 |
| | $C_{LL}$ | 7.00e-03 | 6.38e-01 | 5.865e+03 | 1.02 |

[a]) All models are derived from declustered normal score transformed variables of depth to bedrock $D$ and horizontal distance to nearest outcrop $L$. Practical range, $\beta = \log(0.05)$ for all models.





**Table 4.** Cross-validation results from ordinary kriging (*OK*) and co-kriging (*CK*) with model parameters correponding to cases given in Tab.(3).

| Case | | $M_{AE}$(m) | $S_{AE}$(m) | $F_{AC}$(-) | $M_{SP}$(-) | $S_{SP}$(-) |
|------|------|------|------|------|------|------|
| A | *OK* | 4.44 | 7.72 | 0.16 | 0.39 | 0.098 |
|   | *CK* | 4.52 | 7.58 | 0.40 | 0.37 | 0.125 |
| B | *OK* | 4.36 | 7.57 | 0.65 | 0.34 | 0.062 |
|   | *CK* | 4.33 | 7.51 | 0.62 | 0.38 | 0.077 |
| C | *OK* | 4.35 | 7.46 | 0.62 | 0.39 | 0.098 |
|   | *CK* | 4.31 | 7.40 | 0.57 | 0.44 | 0.116 |
| Case | | $M_{AE}$(m) | $S_{AE}$(m) | $F_{AC}$(-) | $M_{SP}$(-) | $S_{SP}$(-) |
| D | *OK* | 4.43 | 7.71 | 0.66[b] | 0.31 | 0.039 |
|   | *CK* | 4.38 | 7.54 | 0.59 | 0.43 | 0.065 |
| E | *OK* | 4.37 | 7.47 | 0.59 | 0.44 | 0.151 |
|   | *CK* | 4.29[a] | 7.23 | 0.51 | 0.41 | 0.151 |
| F | *OK* | 4.35 | 7.45 | 0.59 | 0.44 | 0.151 |
|   | *CK* | 4.30 | 7.27 | 0.53 | 0.49[c] | 0.173 |

$M_{AE}$ - mean absolute error (25).

$S_{AE}$ - standard deviation of absolute error (26).

$F_{AC}$ - fraction of estimates that fulfill the accuracy criteria (29).

$M_{SP}$ - mean scaled precision (32).

$S_{SP}$ - standard deviation of scaled precision (33).

[a] - lowest mean absolute error.

[b] - highest accuracy.

[c] - highest precision.