# Peer review of "Estimating unconsolidated sediment cover thickness by using horizontal distance to bedrock outcrop as secondary information"

_Hydrology and Earth System Sciences, 2016_

## Referee Comment (RC1) · W.H. Farmer (Referee) · 22 Feb 2017

The author has written an admirably detailed manuscript of the value of kriging and co-kriging to estimate sediment thicknesses in Norway. I am not overly familiar with the content area, but am familiar with the methodology applied. As such, I have, for the most part, restricted my comments to an assessment of methodology. Below I include several comments that may be useful, but, at the editor's discretion, I see no major impediments to eventual publication.

Page 2, line 5: Typographical error.

Page 2, line 24: Throughout the manuscript the term "significant" is used to mean "large". I would strongly reserving "significant" to refer to the result of a quantified statistical test. (Other examples are on page 13, line 28, and page 16, line 27.)

Page 3, line 5: While amusing, I didn't quite follow what was meant by "the big 'Zoo' of different methods".

Page 5, line 14: In my opinion, the removal of 750 sites belongs in the previous paragraph and the final number on line 13 should be revised accordingly.

Page 5, line 15: I found myself debating if this exploratory data analysis was more appropriate as a result. Page 5, line 16: Could the author provide some quantification of the normality of these values? Probability plot correlation coefficients or even probability plots may be useful in highlighting the non-normality.

Page 5, line 22: It is not immediately clear what is meant by "searching windows". Are these related to the windows discussed on page 7, line 10? Further explanation would be useful.

Page 6, line 24: I think "origin" is more common than "origo", but it may be a matter of style.

Page 7, line 20: Given the complexity of kriging methods applied, which almost certainly used computer approximations, it seems a bit odd that the author deferred to probability tables rather than using computer approximations.

Page 7, line 27: Please provide some additional discussion of the implications of not reproducing censored observations. How does this relate to over-estimation of low values seen later?

Page 9, line 12: Why was the exponential semivariogram chosen? Were others considered? Given the steepness and short range in Figure 6C, it appears that a constant model might have been more appropriate for the cross-semivariance.

Page 13, line 28: I find this point on over and under-estimation particularly interesting in light of my own work (DOI: 10.1002/2016WR019129). Please do not take this as a request to pad citations; I mention it only to encourage more discussion. Is this as an effect of model smoothing? The kriging algorithm, by nature, makes predictions as

linear sums of observations. It is thereby expected that extremes might be moderated by less-extreme values, however smally-weighted, in the summation. Perhaps this is worthwhile discussion, perhaps not.

Page 14, line 1: I did not understand this sentence. Furthermore, by what hypothesis test was the difference determined to be significant?

Page 16, line 32: This paragraph borders on advocacy, a style I tend to shy away from in scientific literature. Furthermore, I think it belittles the purpose of this work: The impact of the manuscript is not in "increase[ing] research on data from GRANADA" but rather a strong demonstration of the usefulness of kriging co-kriging for interpolation of soil characteristics. I would support removing this entire paragraph.

Figures 10 and 11: Not being of the region, I though country outlines would be useful here.

Table 3 and 4: What are the definitions of the cases is not clear to me. Perhaps I missed it, but a description should be in the methods section.

I have not completed a complete editorial review of grammar and style. I notice a handful of typographical errors, but I leave these to copy editing.

Finally, thank you for the opportunity to review this manuscript. It represents a well-written application of kriging. I hope my comments have been useful and can be contacted if any additional information would prove useful.

Thank you,

William Farmer

Email: wfarmer@usgs.gov

---

## Author Comment (AC1) · 29 Mar 2017

Thanks a lot for constructive questions and comments! I have worked through all of them, and I have revised the manuscript according to the reply below.

On C1. The purpose of the paper was to explore possible cross-correlations between the thickness of sediments, $D(u)$, and horizontal distance to exposed bedrock, $L(u)$. Cross-correlations might be capitalized later to improve estimates of $D(u)$. This question was tested by using ordinary kriging and co-kriging. Thus, the paper was not intended to describe "the value of kriging and co-kriging", as indicated by Dr. Farmer. Nevertheless, I appreciate this general comment very much.

The typographical error (P2, L5) is corrected, and the term "significant" is substituted with other synonyms as for example "large".

[Figure]

I agree that scientific writing should be as precise as possible. This is a motivations for using mathematics! At the same time there is a lot of useful (common) words that also has a distinct mathematical or statistical meaning. "Expectation, variance, and correlation" are all examples of such words, they have a specific mathematical definition, but they might also be used in another context to express something "significant".

On C2. P3, L5: My intention was not to be amusing, therefore "Zoo" is substituted with "number" in the manuscript. However, a competition is going on out there and the fittest method will survive ("jungle" might therefore be a more adequate metaphor than "Zoo"!). My point is simply that the current study is not a part of the on going competition, I just want to investigate whether the secondary variable (L) might be used to improve the estimates or not.

P5, L14: I do not agree. There are two populations of wells, those located on exposed bedrock, L(u)=0, and those not located on exposed bedrock, L(u)>0. The 750 wells belongs to the second population and should therefore not be described in the previous paragraph. Even though L(u)>0, there are 750 where D(u)=0. These wells are removed in the current study to keep the analysis as simple as possible. It is interesting to notice that number of wells (750) is only a minor fraction (3.7%) of the wells with L(u)>0 (20432). One reason is that the soil cover is very patchy some places, but there is no serious inconsistency between the mapping (and definitions) of D and L.

P5, L15: I think descriptive statistics belongs to exploratory data analysis, and I recognize that others do the same (e.g. Goovaerts et al. 2005, Water Resour.Res., Vol. 41, W07013, doi:10.1029/2004WR003705, 2005). Thus, at this point I suggest no revision of the text.

P5, L16: I've made a (sloppy) normality plot, which indicate that log10(D) and log10(L) did not belong to a Gaussian pdf (c.f. enclosed figure). The main reason for not using a lognormal variable, however, is the (large) estimation error associated to the log-transform. The estimation error of a lognormal variable becomes usually very large

in some (extreme) locations because (by definition) the error of a lognormal variable includes the expected value. This is the main reason why the lognormal transformation has fallen into disuse in science.

P5, L22: No, it is not the same searching window as mentioned on P7, L10. To avoid any confusion I have rewritten the paragraph to: "Mean and standard deviation of $D$ and $L$ as a function of separation distance $h$, is given in Fig.3 for $\Delta h = 20$ m and $\Delta h = 150$ m."

P6, L24: I would prefer "origo" (location of point zero), which I find more specific than "origin", but both terms may work equally well.

P7, L20: I tried both a computer approximation and a analytical expression of the Gaussian pdf, and according to my experience, the analytical expression was more precise especially for the extreme parts of the distribution.

P7, L27: The paragraph ("The back transformation, however, does not reproduce the censored part of the pdf.") is deleted in the revised manuscript. The normal-score transform is done on declustered data, which removed (smoothed) the censored character of the raw data. This is not related to the problem of over-estimation of low D and under-estimation of large D.

P9, L12: The first order exponential model (\alpha = 1 in eq. 14 and 16) is usually referred as the "the exponential semivariogram model". Even though \alpha is close to one, this is not the "chosen" model. The alpha parameter in eq. (14) and (16) may vary between 1 (exponential model) and 2 ("gaussian" model). Cross-validation is also done with $C\_0 \sim= C\_1$, which is similar to a "constant" model (c.f. Tab. 3, case A).

P13, L28: Thanks a lot for your humble reference to Farmer and Vogel, 2016! Quantification of uncertainties is a part of the estimation problem, I totally agree! Over-estimation of (extreme) small values and under-estimation of (extreme) large values is a well known problem for Gaussian least square methods. The purpose of the paper is

not to improve ordinary kriging or co-kriging, but to demonstrate the effect of including a secondary variable (L). Of that reason, I do not intend to elaborate this subject any more in the current manuscript. There are however, made significant progress on the problem of Gaussian and non-Gaussian estimation, and I therefore include references to: Omre and Haldorsen, 1989; Rue et al., 2009; Lindgren and Linström, 2011; Leblois and Creutin, 2013; Ingebrigtsen et al., 2014; 2015.

P14, L1: The paragraph is rewritten to: "In cases with minor difference in absolute error, the estimation results might be ranked according to criteria for estimation accuracy (28) and precision (31)."

P16, L32: Yes, I agree, and the paragraph is rewritten! However, I think the value of public databases on hydrology and environmental databases in general, should be more honored by the scientific society. of that reason I suggest to include the paragraph: "Hence, in this context, the present study is a call to explore public data to obtain important estimates for science and society."

Figure 10 and 11: I decided to include a simple map of Scandinavia, which also indicate the three subsections: Southern Norway; Northern Norway; and the Oslo region. If possible, I would suggest to merge Fig. 10 and Fig. 11.

Table 3 and 4: From my point of view, the presentation of Tab. 3 and 4 does not belong to the method section. Both tables are results of the applied methods. I think the results should be presented as clear as possible without any comments, and interpretation and explanations therefore belongs to the discussion section. Thus, I suggest to keep the text on this point as it is.

With respect to grammar, style and typographical errors, I appreciate all help and corrections! Thanks a lot!

Nils-Otto Kitterød <nils-otto.kitterod@nmbu.no>

[Figure]

Interactive
comment

[Figure]

**Fig. 1.** Standard normalized log10(D) and log10(L) plotted against a Gaussian variable, N(0,1).

---

## Referee Comment (RC2) · P. Sadler (Referee) · 30 May 2017

The title does not clearly reflect the contents of the paper. To a geologist, sediment can crop out. Outcrop does not imply basement. Therefore, I assumed the paper would discuss the relationship of a sub-surface unit thickness to the distance from its own nearest outcrop. The abstract did not correct this false impression. After reading the whole text I suggest that the title should have been: "Estimating unconsolidated sediment cover thickness by using horizontal distance to bedrock outcrop as secondary information." Distance to outcrop could also be termed distance to zero isopach. Had I understood this from the title or abstract I would not have agreed to referee the paper; I have too little first-hand experience with semi-variograms and kriging to be considered an expert referee.

[Figure]

The paper clearly and thoroughly investigates the value of adding distance from the zero isopach to standard kriging of point measurements of the thickness of young sediment cover. It concludes that co-kriging with that additional information reduces the absolute error and improves precision.

The paper tests co-kriging against borehole data for the whole of Norway. The data handling is explicit. The paper provides some standard, straightforward suggestions for culling data from public databases, for avoiding bias due to cell size and grid origin (not origo?), and for mitigating biases introduced by uneven sampling and outlier data. The many equations that aid the understanding of the methods range from trivial formulations for error, mean and standard deviation to more complex kriging matrices that I am not fully competent to check for accuracy. Fortunately, the methods are all supported by references to established literature. The exploratory statistics and the cross-validation are both thorough. The application to a large area like Norway seems contrived, but it serves to demonstrate the utility of adding knowledge of distance from the zero isopach; i.e. adding geologic map information. For more local studies, there would seem to be better non-invasive geophysical controls on the thickness of sedimentary cover; i.e. gravity anomalies. In many regions the public databases would include gravity anomaly maps. For tectonically active regions, the mapped traces of faults would seem to be a complementary source of control for sediment cover thickness.

Although the overall presentation is well-structured and clear, the manuscript contains many grammatical flaws, notably the mismatch of subject and verb (singular and plural), but none of these obscures the meaning. At first, I had difficulty understanding dual use of h in the application of window sizes in delta-h to moment measures of D and L as a function of separation, h. Otherwise, the text was mostly clear on first reading.

Minor flaws by page and line number (page:line):

1:4 tested by comparing

2:3 soil properties . . . have

2:5 one example is

2:14 one of the variables . . . is

2:24 delete"horizontally"

2:33 for minor, write small

3:16 The area . . . is

3:24 drilling is terminated [before reaching basement] because

4:26 erosion [products] were deposited

5:4 There is no mandatory method

5:8 Wells . . . were also

6:26 origin

7:1 origin

7:10 in geostatistical terminology (delete the)

7:22 the data are censored

8:6 the mean . . . and variance . . . were calculated

9:26 which makes

12:26 maps . . . were

13:3 cannot be ruled out

13:6 the data (delete material)

14:28 clustering has

14:31 and then calculation of experimental

15:8 inversely proportional

15:32 analysis indicates

16:10 Despite these uncertainties (delete of)

16:18 despite the significant (delete of)

16:18 experimental data (delete material)

16:27 the number . . . has

16:30 for "and" write "but"

17:8 database . . .was used

17:13 differences were

22:Figure 2 caption lower right and upper left are reversed

---

## Author Comment (AC2) · 30 May 2017

Thank you very much for reviewing the manuscript! I agree that the title would benefit on the two precisions that you suggested. The revised title of the manuscript is then: "Estimating unconsolidated sediment cover thickness by using horizontal distance to bedrock outcrop as secondary information".

To make the abstract clearer and more consistent to the title, I would also suggest to substitute "Sediment thickness (D) . . . " by "Unconsolidated sediment cover thickness (D) above bedrock . . ." (page 1, line 1), and "nearest outcrop (L) . . .", by "nearest bedrock outcrop (L) . . ." (page 1, line 3).

Reply on C1: It is true that the suggested method is "contrived to" large scale estimation, and that the estimation uncertainty usually is too big for small scale deterministic

(or local engineering) purposes. It does not mean, however, that geostatistical estimation is useless for small-scale estimation. The key issue is to minimize the estimation uncertainty, and to obtain that goal all available information should be taken into account. With that respect geophysical prospecting methods, like gravity anomalies or seismic mapping provide of course useful information. The estimation uncertainty is therefore very often a question of costs. For hydrological applications like estimation of storage capacity of water in unconsolidated sediments, it may be necessary with estimates in areas where no measurements are available. In such cases, methods that utilize available information to minimize estimation uncertainty is of great interest.

I recognize that both Sadler and Farmer are a bit disturbed by the concept "grid origo" and suggest to use the term "grid origin". To me the term "grid origo" means the location where the grid coordinates are zero, while the term "grid orgin" alludes more to "where the grid is coming from". The point is that the whole grid has to be moved in order to calculate stable weights. The purpose of moving the "origo" is clearly expressed in the manuscript, and I think it is misleading to say that the "origin" is moved. So, I still prefer to keep the concept "grid origo", even though it may sounds a bit strange for the native English speaking community.

The ambiguity with respect to window-sizes and delta h was also pointed out by Farmer. I suggest a minor revision of the text to avoid this ambiguity (c.f. the reply to Farmer's review).

Reply on C2 and C3: I'm also very grateful for all help with respect to my written English language. The grammatical flaws indicated in C2 and C3, will be corrected in the final manuscript.

---

## Author Comment (AC3) · 5 Jun 2017

Thanks a lot for comments to clarify the terminology on origin vs. origo! I've substituted 'origo' with 'origin' in the revised manuscript.

---

## Author Response (AR1)

Authors final respons to editor.

Thanks a lot for all constructive comments and helping me to publish this work! I have up-loaded a revised version of the manuscript and a revised version of the abstract. All corrections are commented in the reply to the reviewers. If you need a list of corrections (which will be a copy of my reply to the reviewers), please let me know!

Nils-Otto Kitterød

<nils-otto.kitterod@nmbu.no>

---

## Author Response (AR2)

Editor's comment is numbered (1 to 3), and my reply in indent.

1) Add Eq. in front of all references to equations

I've included "Eq." in front of all equation references. I hope the text as it is now, is consistent to the HESS standard. If not, please let me know!

2) Please check the language carefully once more, e.g. Overestimated is one word.

The manuscript has been 'washed' by a native English-speaking person during the review process, but after reading carefully through the manuscript once more I've realized there was still quite a few errors. These mistakes are now corrected. The meaning of the text, however, is not changed. I appreciate very much this opportunity to correct the errors before publishing, and I hope the current version does not have too many language errors. To make the corrections traceable, I've made the following list of corrections:

Page 1, line 14 and 15,
changed from:
"Global warming and natural climate fluctuations give rise to urgent calls for society to quantify impacts on the hydrological cycle."
to:
"Global warming and natural climate fluctuations give rise to urgent calls from water authorities to quantify impacts on the hydrological cycle."

Page 2, line 28,
changed from
"… from a location, Norway, where …"
to:
"… from an area where …"

Page 2, line 33,
canceled:
"…, which imply that they are positively skewed"
(Because it is redundant information.)

Page 4, line 24,
changed from:
"… a few words on the geological setting is necessary."
to:
"… a few words on the geological setting are required."

Page 4, line 25,
changed from:
"… sediments on mainland Norway is …"

to:
"… sediments on mainland Norway are …"

Page 5, line 23,
changed from:
"Mean and standard deviation … is …."
to:
"Mean and standard deviation … are …."

Page 5, line 25 and 26,
changed from:
"…, which are small for minor separation distances, increase to …"
to
"…, which are small for minor separation distances, and which increase to …"

Page 7, line 28,
changed from:
"… the true value are …"
to:
"… the true value is …"

Page 10, line 10,
changed from:
"… the observations: …, where available …"
to:
"… the observations: …, were available …"

Page 10, line 17,
changed from:
"… matrix … denote …"
to:
"… matrix … denotes …"

Page 10, line 17 and 19,
"$k=D,L$" moved to after reference to Eq.(16).

Page 11, line 3,
changed from:
"… can be written (Myers, 1982)."
to:
"… can then be written (Myers, 1982):"

Page 11, line 5,
changed from:
"The total variance is the …"
to:

"Hence, the total variance is equivalent to the …"

Page 11, line 8 to 13:
Slightly rephrased.

Page 11, line 19,
changed from:
"… accuracy … are …"
to:
"… accuracy … is …"

In Page 11 and 12,
I've also removed some parentheses to make the text less clumsy.

Page 12, line ,
changed from:
"The location of boreholes …"
to:
"The GRANADA boreholes used in the current study, …"

Page 12, line 18,
Commas removed after "mean" and before "windows"

Page 12, line 15,
changed from:
"… above (3.1) …"
to:
"…  in section 3.1 …"

Page 13, line 2,
changed from:
"…, the artefact was  …"
to:
"…, the censored character was less obvious …"

Page 13, line 29,
changed from:
"… criteria …"
to:
"… criterion …"

Page 14, line 7,
changed from:
"…  the absolute error …"
to:

"… the mean absolute error …"

Page 14, line 9,
changed from:
"… over estimate…"
to:
"… overestimate …"

Page , line ,
changed from:
"… criteria …"
to:
"… criterion …"

Page 14, line 10,
changed from:
"… the absolute error (10) and the mean of the absolute error (25) …"
to:
"… the mean absolute error (Eq. 25) …"

Page 14, line 14,
changed from:
"… difference in absolute error …"
to:
"… difference in mean absolute error …"

Page 15, line 12,
changed from:
"… do recalculation of …"
to:
"… recalculate the …"

Page 15, line 13,
changed from:
"… semivariograms were …"
to:
"… semivariograms, however, turned out to be …"

Page 15, line 20,
changed from:
"… weights are …"
to:
"… weights were …"

Page 15, line 21,
changed from:

"… (3.1) …"
to:
"… (c.f. section 3.1) …"

Page 15, line 23,
changed from:
"…  semivariograms was …"
to:
"…  semivariograms were …"

Page 16, line 2 and 3,
changed from:
"By similar token, horizontal distance …"
to:
"A similar token was applied in the current study. Here, horizontal distance to outcrop $L(u)$ was evaluated as secondary information to control impact of biased observations of sediment thickness $D(u)$."

Page 16, line 14,
changed from:
"… as done …"
to:
"…  as was done …"

Page 16, line 15,
changed from:
"… OK and CK overestimates …"
to:
"… OK and CK overestimated …"

Page 16, line 15,
changed from:
"This result is …"
to:
"Such results are …"

Page 16, line 33,
changed from:
"…  Norway) is …"
to:
"…  Norway) was …"

Page 17, line 22,
changed from:
"… OK estimates demonstrating …"
to:

"... OK estimates. This result demonstrates ..."

Page 17, line 23,
changed from:
"... an secondary ..."
to:
"...  a secondary ..."

Page 19, line 19, and page 21, line 1.
Minor cosmetic changes in references: added space between family name and first name.

Page 23, line 3 in figure text,
changed from:
"...  distance to nearest ... were ..."
to:
"...  distances to nearest ... were ..."

3) The sentence starting on line 5 in the abstract is a bit confusing. From the text (I think) I understand what you mean, but please clarify here and also later how errors can be similar but accuracy different.

Thank you for making this comment! I understand that this might be a bit confusing, and I've changed the text in the abstract in
page 1, line 5 and 6,
from:
"The analysis showed only minor differences between OK and CK in terms of absolute estimation error, however CK produced more precise results than OK."
to:
"The analysis showed only minor differences between OK and CK with respect to differences between estimation and true values. However, the CK results gave in general less estimation variance compared to the OK results."

I also rephrased slightly
Line 11 and 12:
From:
"Despite of the noisy character, the analysis demonstrates that $L$ can be used as a secondary information to reduce the estimation variance of $D$."
To:
"Despite of the noisy character in the observations, the analysis demonstrated that $L$ can be used as a secondary information to reduce the estimation variance of $D$."

The absolute error (Eq.24) is the difference between estimated value and true value. The mean absolute error (Eq.25) is the average from all cross-validation results. In addition, each single estimate has an estimation variance (Eq.23). This variance is included in the quantity called accuracy (Eq.28) and precision (Eq.31). Thus, accuracy and precision include the 'spreading' (i.e. the variance) of the estimates. Of that reason, the mean absolute error can be similar for two methods, but the precision might be difference. I've read through the manuscript carefully, and from my point of view this should be clear enough. I know I'm a bit biased at this point, but the equations should be quite evident.

Please, let me know if there is anything more you would like me to consider, and thanks again for your careful editor work!